

Airborne particulate matter monitoring in Kenya using calibrated low cost sensors
Francis D. Pope[1]*, Michael Gatari[2], David Ng'ang'a[2], Alexander Poynter[1] and Rhiannon Blake[1]
[1]*School of Geography, Earth and Environmental Sciences, University of Birmingham, Birmingham,*
*United Kingdom, B15 2TT.*
[2]*Institute of Nuclear Science and Technology, University of Nairobi, Nairobi, Kenya,*
*Corresponding author – f.pope@bham.ac.uk*
**Abstract**
East African countries face an increasing threat from poor air quality, stemming from rapid
urbanisation, population growth and a steep rise in fuel use and motorization rates. With few air
quality monitoring systems available, this study provides the much needed high temporal resolution
data to investigate the concentrations of particulate matter (PM) air pollution in Kenya.  Calibrated
low cost optical particle counters (OPCs) were deployed in Kenya in three locations: two in the
capital of Nairobi and one in a rural location in the outskirts of Nanyuki, which is upwind of Nairobi.
The two Nairobi sites consist of an urban background site and a roadside site.  The instruments were
composed of an AlphaSense OPC-N2 optical particle counter (OPC) ran with a raspberry pi low cost
microcomputer, packaged in a weather proof box. Measurements were conducted over a two-
month period (February – March 2017) with an intensive study period when all measurements were
active at all sites lasting two weeks.  When collocated, the three OPC-N2 instruments demonstrated
good inter-instrument precision with a coefficient of variance of 8.8±2.0% in the $PM_{2.5}$ fraction. The
low cost sensors had an absolute PM mass concentration calibration using a collocated gravimetric
measurement at the urban background site in Nairobi.
The mean daily $PM_1$ mass concentration measured at the urban roadside, urban background and
rural background sites were 23.9, 16.1, 8.8 µg m$^{-3}$. The mean daily $PM_{2.5}$ mass concentration
measured at the urban roadside, urban background and rural background sites were 36.6, 24.8, 13.0
µg m$^{-3}$. The mean daily $PM_{10}$ mass concentration measured at the urban roadside, urban background
and rural background sites were 93.7, 53.0, 19.5 µg m$^{-3}$. The urban measurements in Nairobi showed
that particulate matter concentrations regularly exceed WHO guidelines in both the $PM_{10}$ and $PM_{2.5}$
size ranges.   Following a 'Lenschow' type approach we can estimate the urban and roadside
increments that are applicable to Nairobi.  Median urban and roadside increments are 33.1 and 43.3
µg m$^{-3}$ for $PM_{10}$, respectively, the median urban and roadside increments are 7.1 and 18.3 µg m$^{-3}$ for




PM$_{2.5}$, respectively, and the median urban and roadside increments are 4.7 and 12.6 µg m$^{-3}$ for PM$_1$,
respectively. These increments highlight the importance of both the urban and roadside increments
to urban air pollution in Nairobi.
A clear diurnal behaviour in PM mass concentration was observed at both urban sites, which peaks
during the morning and evening Nairobi rush hours; this was consistent with the high measured
roadside increment indicating that vehicular traffic is a dominant source of particulate matter in the
city, accounting for approximately 48.1, 47.5, and 57.2% of the total particulate matter loading in
the PM$_{10}$, PM$_{2.5}$ and PM$_1$ size ranges, respectively. Collocated meteorological measurements at the
urban sites were collected, allowing for an understanding of the location of major sources of
particulate matter at the two sites. The potential problems of using low cost sensors for PM
measurement without gravimetric calibration available at all sites are discussed.
This study shows that calibrated low cost sensors can be used successfully to measure air pollution
in cities like Nairobi. It demonstrates that low cost sensors could be used to create an affordable and
reliable network to monitor air quality in cities.
**1. Introduction**
Recently, the Lancet Commission on pollution and health estimated that in 2015, air pollution led to
the premature deaths of over nine million people globally, and attributed to over one in four deaths
in severely affected countries (Landrigan et al., 2017). Typically, urban air pollution is higher in low
and middle-income countries (LMICs) compared to further developed countries. Hence, the
associated risk of air pollution to health is typically higher in LMICs, with over 92% of global pollution
related deaths occurring in these countries. Within LMICs, health inequalities in urban areas
contribute to an increased exposure to air pollution that faces those that live, work, socialise and
commute to highly urbanised areas which typically have a substantially higher concentration of air
pollutants. Despite the extensive links between air pollutants and human health, environmental
degradation and the economy, air pollution is as of yet still under-researched in many LMICs. Due to
a lack of long term air quality monitoring in many LMICs, the concentrations and sources of air
pollution are poorly understood.
Airborne particulate matter (PM) air pollution is a major environmental risk factor with well-
documented short and long-term effects on human mortality and morbidity (Thurston et al., 2016).
It is known to affect asthma, chronic pulmonary disease (COPD), pulmonary fibrosis, cancer, type-2
diabetes, neurodegenerative diseases, obesity and other conditions (Ferranti et al., 2017). The size
of PM is correlated with their health impacts, with smaller particles typically having more significant
health implications (Meng et al., 2013). PM$_1$, PM$_{2.5}$ and PM$_{10}$ are particulate matter with


aerodynamic diameters less than 1, 2.5 and 10 μm, respectively (Seinfeld and Pandis, 2016). The
World Health Organization (WHO) recommends that $PM_{2.5}$ and $PM_{10}$ daily mass concentrations
should not exceed 25 and 50 μg/m$^3$, respectively; and that annual mass concentrations do not
exceed 10 and 20 μg/m$^3$, respectively (WHO, 2006). At present, the WHO or other regulatory bodies
do not provide recommendations of the mass concentrations of $PM_1$. $PM_1$ can remain suspended in
air for much longer than coarser particulate matter, as well as penetrating deeper into the lungs
leading to local pulmonary, systematic inflammation (Pateraki et al., 2014). Due to the smaller size,
$PM_1$ has a higher surface to mass ratio, containing a harmful amount of potentially toxic
anthropogenic constituents which could lead to health impacts such as respiratory disease, heart
disease and lung cancer (Trippetta et al., 2016). Many studies still focus on $PM_{10}$ and $PM_{2.5}$ even
though smaller particulates pose greater health impacts (Tsiouri et al., 2015). Beyond $PM_1$, ultra-fine
particles (<100 nm) are of such a small size they can be translocated to the central nervous system
via the blood to brain barrier or the olfactory bulb. There are no air quality regulations of $PM_1$ or
ultra-fine particles due to the paucity of data either within environmental science or public health.
Worldwide, road traffic is a dominant source of urban PM accounting for 5-80% of PM mass, with
the precise amount being dependent upon several factors including time, location, and vehicle fleet,
as reviewed by Pant and Harrison (2013). Vehicle derived PM is directly associated with negative
health outcomes (Fan et al., 2006;HEI, 2010). Emissions are due both to exhaust pipe emissions and
non-exhaust pipe emissions. Exhaust emissions result from the combustion of fuel, predominantly
petrol and diesel, and oil and other lubricants.  Non-exhaust emissions come either from the
resuspension of road dust through wind or vehicle induced wind shear, or from the wear and tear of
vehicle parts including the brakes, tyres and clutch.  Resuspension of dust is particularly important
on non-paved roads of which there are an abundance in Nairobi. Typically, non-exhaust emissions
are in the coarse PM size fraction (PM in the size range 2.5-10 μm aerodynamic diameter), whereas
exhaust emissions are in the fine PM size fraction ($PM_{2.5}$) (Thorpe et al., 2007;Kam et al., 2012).
However, it is noted that the papers which reference vehicle PM size distributions according to the
emission of non-exhaust sources have typically been conducted in either the US or European studies
and not in Nairobi or Africa, where non-paved road sources represent a much higher fraction of road
surface type. The precise size of vehicular derived PM is dependent on several factors: vehicle fleet
characteristics (e.g. weight and size), road type and level of maintenance and meteorological
conditions (Beddows et al., 2009;Hays et al., 2011).
In many LMIC cities, urbanization, population, fuel use and motorization rates are all increasing
rapidly and increases in air pollution are associated with these trends (Mitlin and Satterhwaite,



2013;Ochieng et al., 2017). In particular, vehicular traffic is fast on the rise, with associated
congestion on the road networks, which can contribute as much as 90% of air pollution in urban
environments (UNEP, 2005). Nairobi is the capital city of Kenya and is showing these trends. In
particular, the city population has increased dramatically, since 1999 to 2015 it has risen by 83%, and
is projected to increase to 7.14 million by 2030 (Rajé et al., 2017). Similarly, motorization rates are
increasing, between 2008 and 2012, the number of motor- and auto-cycles in Kenya grew by 368%
with the number of overall registered vehicles increasing by 77% (Rajé et al., 2017). Considering this
extensive increase in the vehicle fleet, limited roadway infrastructure and high congestion within the
city, pollution hotspots are created leading to personal exposure levels much higher than that
encountered throughout the rest of the city (van Vliet and Kinney, 2007).
To be able to reduce air pollution, you first have to be able to measure it. Many LMIC countries have
insufficient monitoring networks through which to measure air quality. In particular, long term high
resolution data is required for such cities which are vulnerable to air pollution. Nairobi is in the
vanguard of air pollution measurements for Sub-Saharan Africa but lacks continuous long term
calibrated measurements of PM and other air pollutants (Petkova et al., 2013). A discussion of the
relevant measurements in Nairobi is given in the next section. One of the constraints to making
measurements is the high cost of research grade air quality monitoring equipment with appropriate
calibration and certification. Low cost sensors offer the potential for dramatically reducing
equipment costs by orders of magnitude, making the monitoring of air quality more accessible and
attainable in LMIC countries (Lewis et al., 2016;Rai et al., 2017).
In this paper, the use of low cost sensors for measurement of $PM_1$, $PM_{2.5}$ and $PM_{10}$ in Nairobi is
detailed. We have previously assessed the same low cost sensors in the UK (Crilley et al., 2018). The
sensors are calibrated using a standardised gravimetric approach. PM is measured in three locations:
an urban roadside site, an urban background site and a rural background site. Comparison of
simultaneous measurements at the three sites allows for the estimation of an urban increment and
roadside increment in PM following a 'Lenschow' type approach (Lenschow et al., 2001). The
variation of measured PM with measured meteorological data is also discussed. Finally, we discuss
the implications of using low cost sensors in Nairobi and LMIC countries in general.
**2. Previous PM measurements in Nairobi**
In general, long term air quality monitoring in Sub-Saharan Africa (SSA) is rare. Correspondingly,
there are only limited PM data sets for East African urban areas; where data does exist estimated
concentrations for $PM_{2.5}$ concentrations are typically ca. 100 μg/m$^3$ compared to <20 μg/m$^3$ in most



European and North American cities (Brauer et al., 2012). This indicates that urban PM air pollution
in East Africa is of a significant health concern.
In Nairobi, there have been numerous short term measurements of PM over the last decade (Brauer
et al., 2012;Kinney et al., 2011;Ngo et al., 2015;Egondi et al., 2016;Gaita et al., 2016) with only one
long term continuous measurement (Gaita et al., 2014). To date, most measurements have used
gravimetric measurement methodologies to record PM mass concentration in the $PM_{2.5}$ and $PM_{10}$
size fractions. Most measurements indicate PM concentrations in Nairobi regularly exceed the WHO
guidelines. At present, there is only one publication in the scientific literature describing the use of
low cost sensors in the measurement of PM (de Souzza et al., 2017) which monitored air quality in
Nairobi at six sites from May 2016 to January 2017. Using AlphaSense OPC-N2's, the authors
measured $PM_1$, $PM_{2.5}$ and $PM_{10}$ as well as $NO_2$, NO and $SO_2$.
The study collected PM concentrations at six schools within Nairobi. It reported a $PM_{2.5}$
concentration range between 11 and 21 $\mu g/m^3$, and a range of 26 to 59 $\mu g/m^3$ for PM10. The PM
concentrations measured during the de Souzza study are noticeably lower than of this study for both
size fractions. It is worthy of note that the de Souzza study collected measurements from May 2016
to January 2017, whereas this campaign took place from February to April 2017; the local
meteorology may have influenced the discrepancies seen in both recorded PM concentrations.
Additionally, the study did not calibrate the monitors, which leads to questions about absolute
concentrations and interference from other environmental dependencies (Lewis and Edwards,
2016). The collected data from the study appeared noisy, with the authors stating they could not
separate the signal from the noise without having access to an air quality measuring reference
instrument (they recorded peaks at over 1000 $\mu g/m^3$). Despite the limitations, it provides a useful
comparison to this calibrated study.
The paucity of long term calibrated measurements has hindered the understanding of long term
trends and the influence of seasonal variations in meteorology and other factors. Most published
data provide daily averages of PM mass; the lack of higher temporal resolution data precludes the
generation of diurnal data which can be useful for identifying individual sources of PM, in particular,
vehicular PM which typically tracks traffic and hence peaks during rush hours.
The longest record of PM concentration in Nairobi is detailed in Gaita et al. (2014). In this work, the
authors performed daily measurements of $PM_{2.5}$ at an urban background and suburban site over a
two-year period from May 2008 to April 2010 using polycarbonate filters in cyclone sampler (Casella



from Bedford, UK). They reported a concentration range of 3 µg/m$^3$ to 53 µg/m$^3$ at the urban
background site, with an overall mean of 21 µg/m$^3$ which exceeds the annual WHO limit of 10 µg/m$^3$
by a factor of two. The average concentrations of PM$_{2.5}$ at both sites were found to be 21 ± 9.5 and
13 ± 7.3 µg/ m$^{-3}$, respectively.  Chemical composition measurements of the filter samples allowed
source apportionment, via positive matrix factorization, to be carried out. The analysis suggested
that five major source factors contribute to Nairobi PM$_{2.5}$: traffic, mineral dust, industry, combustion
and a mixed factor. The dominant source factors were mineral dust and traffic which accounted for
74% of the particle mass.
As an update to this study, Gaita et al. (2016) conducted a study on the characterization and size-
fractionation of particulate matter and deposition fraction in the human respiratory system in
Nairobi using measurements taken in August and September 2007, obtained at the University of
Nairobi site. Based on the findings, the concentration levels of airborne particulate matter sampled
at the urban background site during the period was found to range between 1 µg/m$^3$ and 78 µg/m$^3$.
The average PM$_{2.5}$ concentration at the site over the entire sampling period was 9.8±8.5 µg/m$^3$.
A densely populated urban area with associated heavy local traffic within Nairobi largely contributes
to the city's air pollution build up. Kinney et al., (2011) investigated the impact of vehicular
emissions in Nairobi on the concentration of PM$_{2.5}$, observing a substantial range between 58 µg/m$^3$
and 98 µg/m$^3$ across an 11-hour personal exposure along busy roadways and roundabouts. The
range could be estimated to be between 45 and 85 µg/m$^3$ for a 24 h sampling due to pollutant
dispersion at night. In addition, the study reported a decrease in horizontal dispersion
measurements of PM$_{2.5}$ from 128.7 µg/m$^3$ to 18.7 µg/m$^3$ over 100 m downwind of a major
intersection in Nairobi. A vertical dispersion from a street level to a third-floor rooftop in the Central
Business District (CBD) showed a decrease in PM$_{2.5}$ concentration from 119.5 µg/m$^3$ to 42.8 µg/m$^3$.
This study clearly highlights that the PM concentration in Nairobi varies considerably over both time
and space, which has significant implications for human exposure, see discussion.
Another study by Ngo et al., (2015) affirmed the contribution of anthropogenic activities on the
quality of air in Nairobi. In their study, Teflon filters in PM$_{2.5}$ samplers (BGI model 400) were used
between 2$^{nd}$ August and 18$^{th}$ August 2011 and high concentrations of PM$_{2.5}$ exposure levels among
different groups in Nairobi were reported. According to the study, bus drivers in Nairobi city were
exposed to about 103 µg/m$^3$ while those in informal settlements, such as Mathare, reporting
exposure levels of about 62.7 µg/m$^3$, an indication that urgent measures needed to be taken to
mitigate the impact of air pollution in the city.





The severity of air pollution in urban centres in SSA is typically even higher in the informal
settlements (slums), where acute respiratory tract infections and bronchitis are among the most
frequent medical diagnoses (Gulis et al., 2004). Egondi et al., (2016), in their study on air pollution in
two informal settlements in Nairobi: Korogocho and Viwandani, reported higher concentration levels
of $PM_{2.5}$ in the two slums. Optical counters (TSI DustTrak II model 3530) were used in the study and
observed average concentration levels of $PM_{2.5}$ in Korogocho slum, lying west of Dandora, Nairobi's
biggest dumping ground, were the highest at 166 $\mu g/m^3$ and Viwandani, situated North of Nairobi
recorded 67 $\mu g/m^3$.
**3. Methodology**
**3.1. Site locations**
This study utilised three field sites in Kenya, see Figure 1. Two sites were in Nairobi which is the
capital of Kenya, covering an area of ca. 696 sq. kilometres and home to approximately 3.5 million
residents according to a World Population review conducted in 2016 (Kenya Population, 2016),
making it the second most populated city in East Africa after Dar es Salaam, Tanzania. In addition to
Nairobi's longstanding popularity as a travel destination, due to its safari and other holiday resorts,
the city also acts as East Africa's diplomatic, financial and communication capital (Rajé et al., 2017).
Its geographical location is at approximately 1.29° S and 36.82° E. The highest elevation point in the
city is at an altitude of 1663m above the ground. As discussed in the introduction, Nairobi is
undergoing rapid increases in population and motorization both of which will likely lead to greater
PM pollution in the absence of any efforts of mitigation against the pollution. Other significant
infrastructure projects such as major road building are currently being undertaken, which will also
likely lead to increased PM loadings.  Within Nairobi, the two field sites represent an urban
background location and an urban roadside location.  The other site, a rural background site is
located on the outskirts of Nanyuki, a town that is located at an approximate aerial distance of 147
km to the north (NNE) of Nairobi and 240 km by road. The sensor boxes were placed in locations free
from obstacles, at the three measurement sites, allowing for 360 degrees of air flow.
**Site 1: American Wing, University of Nairobi, Nairobi (urban background site)**
The first site for data collection in Nairobi was at the American Wing building located in the
University of Nairobi, standing at an elevation of 17 m above ground level. Air flow at the site was
free from any obstruction as the OPC's were located at an elevated point above the ground. The



nearest road is Harry Thuku road which has very few on-road vehicles (no heavy trucks) and it leads
to Fairmont Norfolk and Boulevard hotels, and Kijabe Street.  Its level is far below the site and the
only influence from the few vehicles and the city is highly diluted and dispersed pollutants (Kinney,
et al., 2011) in regional air mass.
**Site 2: Tom Mboya Street, Fire station, Nairobi (urban roadside)**
The second collection site in Nairobi was at the fire station, which is located within the CBD in the
city. Unlike the American Wing site, the area around the Fire Station is characterized by high traffic
flow which includes common public transport vans, locally known by the name "Matatus". It is on an
urban street canyon, on a street where smoking diesel vans are frequent and is exposed to urban
heat Island effects.  It is also in the neighbourhood of vertical dispersion measurement site of $PM_{2.5}$
used by Kinney et al. (2011). The monitor was positioned at a height of approximately 5 m.
**Site 3: Nanyuki town (rural background)**
The third site chosen was on the outskirts of Nanyuki town, an administrative town in Laikipia
County which is located to the North West of Mt. Kenya. The town is positioned at the Equator at
approximately 1.28° S and 36.01° E.  The highest point in Nanyuki is at an elevation of 2000 m above
ground level. The town is home to approximately 50,000 people as per the last census conducted
(KNBS, 2015). The Nanyuki region has a hinterland of significant agricultural cultivation, forest and
considerable grazing activities (Gatari et al., 2005). The OPC was hung about 4 m above ground level
thus exposing it to free regional air mass in an area of minimal local influence.



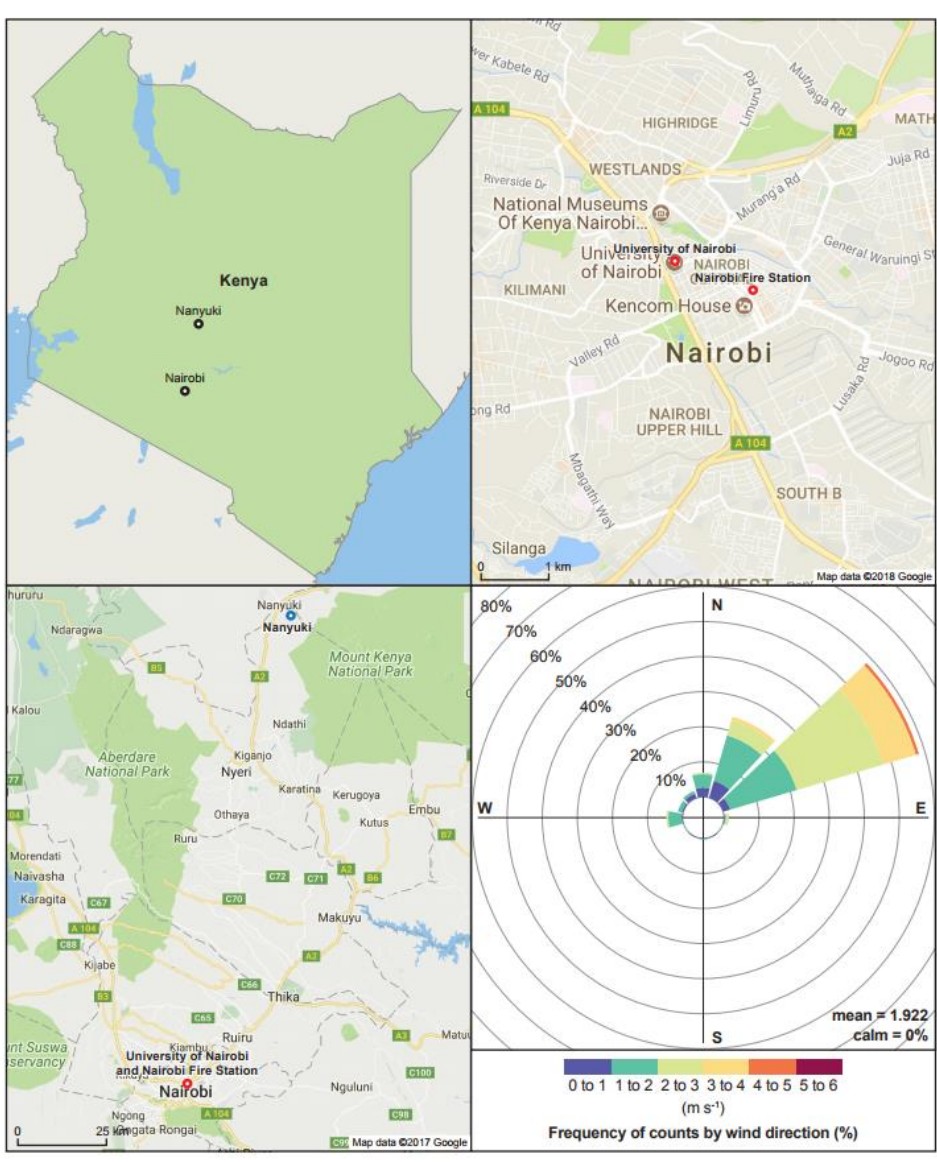

**Figure 1. Locations of data collection sites and wind rose for the urban background location. Top left panel and bottom left panel: locations of Nairobi and Nanyuki in Kenya. Top right panel: shows relative locations of urban background location (University of Nairobi, American Wing) and urban roadside location (Nairobi fire station). Bottom right panel: wind rose generated from data collected at the urban background location during the measurement campaign.**



**3.2. PM Measurement Equipment**
Small low cost optical particle sensors (AlphaSense, OPC-N2, firmware version 18) were used to
measure PM concentrations.  The AlphaSense OPC-N2 sensors are henceforth referred to as OPC-N2.
The OPC-N2 is a miniaturized OPC which has dimensions of 75×60×65 mm and weighs under 105 g.
The unit cost of an OPC-N2 is approximately 250 GBP or 25000 KeS, hence it is significantly cheaper
than reference optical particle counter instruments which cost approximately 30-50 times as much.
Reference grade gravimetric instruments can cost even more. The lower cost of the OPC-N2,
provided the opportunity for measurements at multiple sites simultaneously. It measures particles in
the reported size range of 0.38 to 17 μm across 16 size bins, with a maximum particle count of
10,000 per second. The particle number concentration is converted by on-board factory calibration
to PM concentrations according to European Standard EN481 (OPC-N2 manual).
The assumed density for all particle sizes is 1.65 g/cm$^3$ and no special weighting is placed on any
particular bin size. However, the manual for the OPC states "an additional weighting is applied on
units with Firmware 18 or higher to account for under counting at low particle sizes and the effect of
carbon particles in urban air so that the output matches collocated reference detectors."
The lower cut off for particle size observed by the OPC is 380 nm and hence a large proportion of all
particles are not observed by the OPC due to the particle number being dominated by the smallest
particle sizes (Seinfield and Sypyros, 2016). Ultrafine particles (particles of aerodynamic diameter
<100 nm) were therefore not measured. However, the interest of the study was particulate mass
which is dominated by particle sizes that were measured.
The sensors had their data logged using Raspberry Pi 3 minicomputers.  The Python script used to
run the OPC-N2 on the Raspberry Pi 3 is discussed and provided in Crilley et al. (2018) and makes use
of the py-opc python library for operating the OPC-N2 written by Hagan (2017). Together, the OPC-
N2, minicomputer and accompanying wires and tubing were placed in bespoke weather resistant
housing (dimensions ca. 30*20*10 cm).  Power for both the OPC-N2 and minicomputer were
provided by mains power.
The OPC-N2 sensors are factory calibrated to measure PM mass concentrations representative of
the UK. However, in our previous study, (Crilley et al., 2018) we demonstrated that in situ calibration
of the sensors is required for the correct measurement of PM mass concentrations at urban
background sites in Birmingham, UK.  The calibration in the Crilley et al. (2018) study involved both



scaling and a relative humidity (RH) dependent term for when the RH is greater than approximately

2 85%.

The mass concentrations from the OPC-N2 devices, in the $PM_1$, $PM_{2.5}$ and $PM_{10}$ size bins were
recorded in time intervals of 10 s. For the subsequent analysis, the mass concentration data were
aggregated into 1 h time-bins using the mean average. In time periods which contained missing data,
the mean average of the available data was aggregated. All data manipulations were performed
using R (version 3.4.1), and the openair project package for R was used extensively for data
visualization (Carslaw and Ropkins, 2012).
**3.3. Meteorological station**
The local meteorology for Nairobi was measured at the same location as the urban background site
using a Vaisala instrument (WXT510) with the following variables measured: wind speed, wind
direction, temperature, relative humidity, relative humidity, barometric pressure, and rainfall with
an instrument temporal resolution of five minutes. The meteorology measured parameters were in
good agreement with other local measurements such as those observed at Jomo Kenyatta
International Airport (JKIA), which is approximately at an aerial distance of 10 km. The proximity of
the meteorological station at the urban background site to the urban roadside makes the
meteorological data appropriate for both sites. The data was collected at the urban background site
from the 2$^{nd}$ of February to the 6$^{th}$ of April 2017, covering the duration of the PM measurements.
**3.4. OPC-N2 gravimetric mass calibration**
The OPC-N2 mass concentrations were calibrated using gravimetric measurements of $PM_{2.5}$ and
$PM_{10}$. The gravimetric calibration measurement was carried out on the 9$^{th}$ February 2017 for 24 h. A
collocation measurement of the OPC and an Anderson dichotomous impactor (Sierra Instruments
Inc., USA) was set up, on the only possible date, at the background site. The impactor collected $PM_{2.5}$
and $PM_{10-2.5}$ particles on Teflon filters (diameter = 37 mm, pore size = 2 μm) at a total flow rate of 1
$m^3 h^{-1}$. $PM_{10}$ is therefore the sum of the two size fractions ($PM_{2.5}$ + $PM_{10-2.5}$). The chosen sample day
was rain free and had similar temperature and RH profiles compared to the rest of the OPC sampling
campaign. The filters were weighed using a mass balance before and after particulate matter
collection. The observed 24 h average mass concentrations of $PM_{2.5}$ and $PM_{10}$ from the impactor
were 27.6 ± 6.8 and 51.8 ± 10.3 μg m$^{-3}$, respectively, while those recorded from the OPC 16.9 and
30.6 μg m$^{-3}$, respectively. The uncertainty in gravimetric concentrations was estimated from the
instrument (10%), sampling (7%) and weighing (25%) errors and that of the OPC data was the



standard deviation. Hence, the observed scaling factors between the OPC derived masses and
gravimetric analysis were 1.70 and 1.63 for $PM_{10}$ and $PM_{2.5}$, respectively. These factors are different
to that observed in Crilley et al. (2017) which performed a similar gravimetric calibration procedure
with the OPC-N2 measuring PM at an urban roadside sites in the UK. The discrepancies in scaling
factors are likely due to differences in average particle densities observed in Kenya compared to that
observed in the UK, and also the typical RH measured in Nairobi compared to the UK measurements
(see discussion in next section). In particular, Nairobi PM has been shown to have a high percentage
of mineral dust which typically has a high density, with Gaita et al. (2014) showing the annual
average composition of $PM_{2.5}$ being composed of 35% mineral dust which originates from unpaved
roads and wind-blown dust during the dry seasons. The gravimetric analysis did not allow for the
calibration of the $PM_1$ mass concentrations because a filter sample was not generated for the
fraction of PM in this size range. Hence, the $PM_1$ size fraction calibration uses the same calibration
factor derived for the $PM_{2.5}$ size fraction.
The gravimetric calibration was carried out at the urban background field location, for the three
OPC-N2s which were subsequently used in the measurement campaign at the three field sites.
Hence, the calibration was most appropriate for the urban background site. Whilst the urban
roadside site is in close proximity to the urban background site, the roadside site is more influenced
by traffic related PM, hence, the average particle density at the roadside site is likely different to the
urban background site. Likewise, the rural background site is likely to be far more influenced by
mineral dust than the two urban sites. Hence the gravimetric calibration at the urban background
sites only provides an estimate calibration for the urban roadside and rural background sites.
Only one gravimetric calibration was carried out during the study period due to the lack of resource
for further calibrations. If the PM composition varied significantly over the study period, then the
true calibration factor will also change. Hence, the calibration factor used should be treated as an
estimate for the whole study period because changes in PM composition lead to changes in particle
refractive index, and therefore, the scattering pattern which is measured by the OPC to estimate
particle size. Changes in particle density, due to compositional changes, also affects the particle mass
calculated from the particle size. It is noted, for future studies it would be beneficial to have multiple
gravimetric calibration points to check for continuing accuracy of the OPC-N2 sensors throughout
the campaign.



### 3.5. Measured particle mass dependence on relative humidity

As detailed in Crilley et al. (2018), under UK conditions, the OPC-N2 device is sensitive to variations in RH when the RH exceeds ca. 85%. Crilley et al. (2018) suggest the RH dependence is due to the hygroscopic properties of particles that result in significant water mass being taken up by PM at high RH. This hygroscopic dependence can be modelled using a calibration that uses the κ-Kohler parameterization of aerosol hygroscopicity (Petters and Kreidenweis, 2008). The average κ parameter values for the surface of the Earth in Africa (κ = 0.15±0.12) are lower than for Europe (κ = 0.36±0.16), signifying that the rural background hygroscopicity is much less in Africa compared to Europe (Pringle et al., 2010). It is noted that composition of urban PM will have different hygroscopic properties to the average rural background. However, PM derived from urban emissions are likely to be less hygroscopic than rural PM; therefore, the rural estimates provide a useful upper estimate of particle hygroscopicity in urban centres. All locations used in the study period typically have RH less than the 85% threshold. However, it is noted that the RH dependent measurements shown in Crilley et al. (2018) were performed in the UK whereas these measurements were performed in Kenya. There may be significant differences between aerosol compositions, and hence hygroscopicities, in these two countries albeit both urban areas (Birmingham and Nairobi) will have significant vehicular influence. Measurements of RH at the Kenyan urban background site show that RH was only equal to or greater than 85% less than 1% of the time. Furthermore, there is no significant dependence of either the observed $PM_{2.5}$ or $PM_{10}$ mass concentration upon RH (see supplementary figures 1a and 1b), this is consistent with low hygroscopicity aerosols. The measurement period of work reported in this paper was in the Kenyan dry season with very few rain events, it is noted that if low cost sensors are to be used in the wet season in Kenya then the RH will likely be greater than 85% during significant periods and the hygroscopicity effect will likely need to be accounted for to obtain good measurements.

### 4. Results

### 4.1. Site meteorology

Figure 1d provides the wind rose for the measurement period and Table 1 provides the statistical summary data for the measured meteorological variables during the study period. The wind came predominantly from the northeast with a mean average wind speed of 1.9 m/s. The measurement period was largely dry but there were rain events on the following days: 17[th], 19[th] and 24[th] February, and 17[th] and 22[nd] of March, see grey shaded rectangles in Figure 3. Air mass back trajectory analysis using HYSPLIT confirms that the air masses arriving in Nairobi, during the measurement period, came from the northeast (Stein et al., 2015). It is noted, the Nanyuki rural background field site is located



north to northeast of Nairobi and hence is a sensible choice for the measurement of the rural
aerosol loading arriving in Nairobi. The temperature and relative humidity time series data from the
urban background site is shown in supplementary figure S2.
**Table 1  Summary meteorological data for the urban background monitoring site in Nairobi (2nd**
**February – 23rd March 2017)**

|  | Wind speed (m/s) | Pressure (mbar at 1680 m) | Temperature (°C) | Relative humidity (%) |
|---|---|---|---|---|
| Minimum | 0.1 | 827.3 | 15.2 | 15.0 |
| 1st Quartile | 1.0 | 831.4 | 18.9 | 37.0 |
| Median | 1.6 | 832.4 | 21.5 | 51.0 |
| Mean | 1.9 | 832.4 | 22.1 | 51.4 |
| 3rd Quartile | 2.5 | 833.4 | 25.2 | 66.0 |
| Maximum | 10.5 | 836.4 | 30.7 | 90.0 |

**4.2. Particulate matter measurement**
PM data was collected at the three sampling sites over the time period inclusive of 02/02/2017 and
24/03/2017. Figure 2 provides the time series data for the $PM_{10}$, $PM_{2.5}$ and $PM_1$ data over the whole
measurement campaign. Gaps in data at specific sites are either due to the colocation of two or all
three instruments at one site for cross calibration purposes, due to power failure requiring
instrument restart or OPC malfunctioning.
The inter OPC-N2 precision was measured once during the campaign by co-locating the three
instruments at the urban background site for 3 days for side by side sampling. The three instrument
colocation was carried out during at the start of the campaign (16/02/2017 – 18/02/2017). Two
OPCs were collocated together at the urban roadside site near the end of the campaign (04/03/2017
– 27/03/2017). All instruments gave very similar readings during both co-location periods, the inter-
instrument precision gave a coefficient of variance of 8.8±2.0% in the $PM_{2.5}$ fraction, with no
degradation in inter instrument precision observed over the sampling period. This coefficient of
variance is better than observed in Crilley et al. (2018) but this is expected because of the lower RH
conditions in Nairobi (see later discussion).





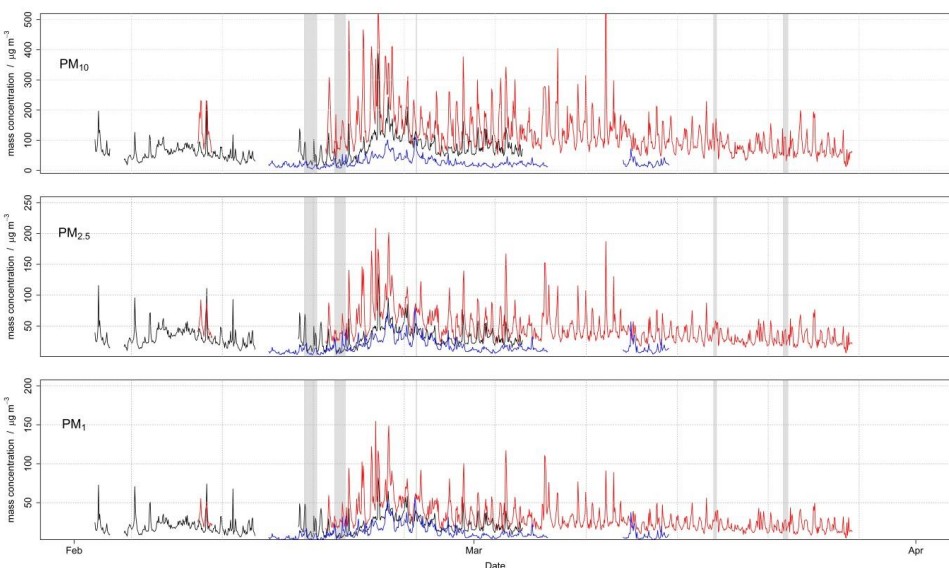

**Figure 2** Hourly time series data showing $PM_{10}$, $PM_{2.5}$ and $PM_1$ mass concentrations at the three
study locations.  Red line = urban roadside, black line = urban background and blue line = rural
background. Where multiple OPC-N2 devices were measuring in the same location at the same time,
the average is provided. The grey shading represents rain events as measured at the urban
background location.
Continuous monitoring at all three sites was achieved for a fortnight in the period 18/02/2017 to
04/03/2017.  This period will henceforth be referred to as the intensive period, whereas, the total
measurement campaign will be referred to as the campaign period. The number of monitoring days
for the urban roadside, urban background and rural background monitoring sites during the
campaign period were 40, 29 and 25 days, respectively.
Table 2 provides the average $PM_1$, $PM_{2.5}$ and $PM_{10}$ mass concentrations observed at the three sites
during the campaign period.   An identical table for the intensive period is included in the
supplementary material, see Table S1. The percentage of daily exceedances of daily $PM_{2.5}$ and $PM_{10}$
as per WHO guidelines are also provided, however, to date there is no set guidelines of $PM_1$.  All
measurement sites exceeded the WHO daily guidelines for both $PM_{2.5}$ and $PM_{10}$ for some of the days
sampled. The urban roadside site exceeds the WHO guidelines on most days (85% for $PM_{2.5}$ and 90%
for $PM_{10}$). Furthermore, on many days (13% for $PM_{2.5}$ and 40% for $PM_{10}$) the urban roadside site
exceeds the WHO guidelines by at least twice as much. The urban background site has fewer
exceedances, compared to the urban roadside site, with daily exceedances occurring approximately



one third of the time.  The urban background site is at an elevated position, which largely removes
the direct influence of local sources of PM pollution.  As such, it can be assumed that the PM mass
concentrations observed at this location represent a lower limit for the ground level PM
concentrations throughout Nairobi, since most PM emissions will be due to ground level sources
such as vehicle emissions, fires, local industry and others.  The rural background site has no daily
exceedances in the $PM_{10}$ size fraction but exceeds the $PM_{2.5}$ guidelines 12% of the time.
During the two-week intensive campaign, there was a period of elevated PM mass concentration
observed in $PM_1$, $PM_{2.5}$ and $PM_{10}$ size fractions centred around the 23[rd] February. The elevated PM
was observed in all three sites; therefore, it likely represents a long-range pollution event.
Correspondingly, the average PM mass concentrations and percentage of WHO exceedances are
higher during the intensive period compared to the whole measurement campaign, see Table S1.
**Table 2** Mean average PM mass concentrations ($PM_1$, $PM_{2.5}$ and $PM_{10}$) and daily exceedances of the
WHO PM guidelines ($PM_{2.5}$ and $PM_{10}$) observed at the three measurement sites during the campaign
period.  [1]WHO guidelines for daily $PM_{10}$ and $PM_{2.5}$ are 50 and 25 $\mu g/m^3$, respectively

| Measurement location | Measurement days (number) | Average $PM_1$ mass concentration ($\mu g/m^3$) | Average $PM_{2.5}$ mass concentration ($\mu g/m^3$) | Average $PM_{10}$ mass concentration ($\mu g/m^3$) | % daily $PM_{2.5}$ exceedances[1] | % daily $PM_{10}$ exceedances[1] |
|---|---|---|---|---|---|---|
| Urban background | 29 | 16.1 | 24.8 | 53.0 | 31.6 | 39.5 |
| Urban roadside | 40 | 23.9 | 36.6 | 93.7 | 85.0 | 90.0 |
| Rural background | 25 | 8.8 | 13.0 | 19.5 | 12.0 | 0.0 |

Whilst there is insufficient temporal data to provide a yearly average value for $PM_{2.5}$ and $PM_{10}$ mass
concentrations for the three sites, the annual average values can be estimated from the data set
using the average values provided in Table 2. These values are likely to be upper estimates for the
yearly values because the measurements were obtained in period with little precipitation, thereby
minimizing the degree of wet deposition of the PM. For instance, Gaita et al. (2014) showed that
Nairobi's short rainy season (typically October – December) suppresses PM concentrations at the
urban background site by approximately 50%.  Notwithstanding the seasonal rain consideration, the



average PM mass concentration observed in this study suggests that that the WHO
recommendations for annual $PM_{2.5}$ and $PM_{10}$ are likely exceeded at both the urban background and
urban roadside locations.  For the urban background site, the measured average $PM_{2.5}$ and $PM_{10}$
mass concentrations exceed the annual WHO recommendations by factors of 2.5 and 2.7,
respectively. Whereas for the urban roadside site they exceed recommendations by 3.7 and 4.7,
respectively. These significant exceedances for both the urban roadside and urban background sites
suggests that most of Nairobi's population will be subjected to outdoor air pollution far in excess of
the WHO recommendations for annual exposure. Figure 3 provides the box and whisker plots for the
hourly averaged $PM_{2.5}$ and $PM_{10}$ data for the three measurement sites, highlighting the proportion of
the days which exceed the WHO annual and daily recommendations.





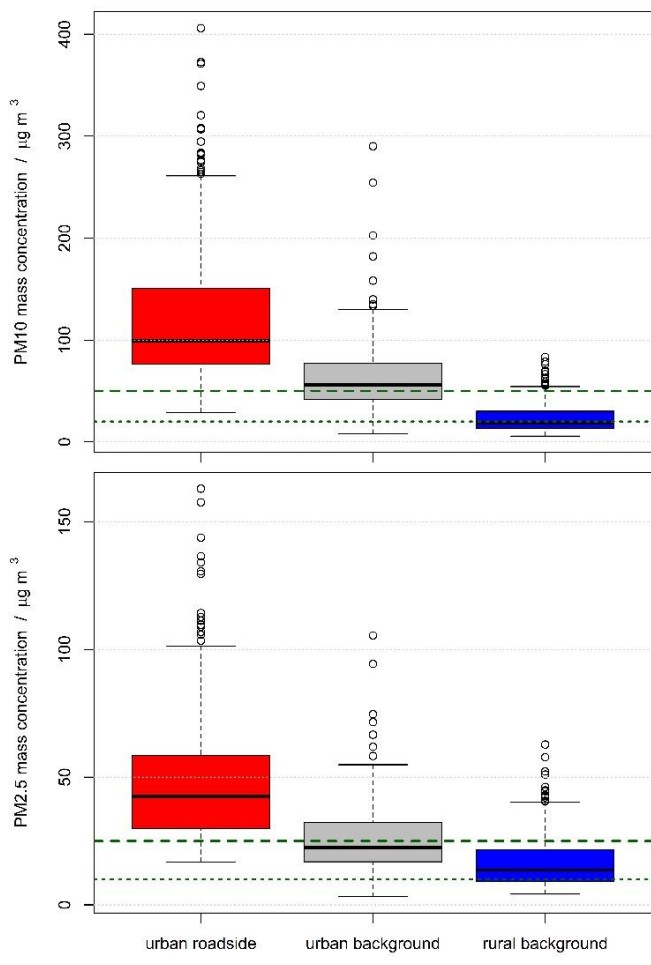

2  **Figure 3** Box and whisker plots of the hourly averaged PM$_{2.5}$ and PM$_{10}$ mass concentrations

3  measured at the three sites. The green dashed and dotted lines represents the WHO recommended

4  annual and daily limits, respectively



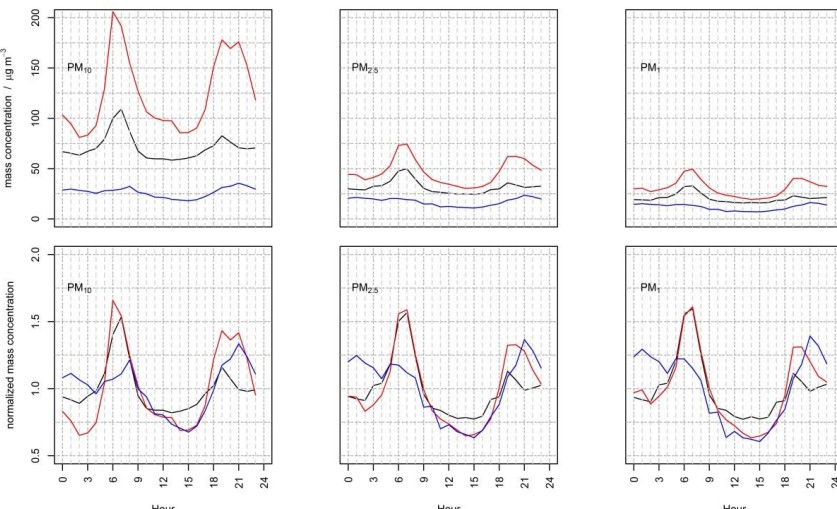

**Figure 4** Diurnal variation in $PM_{10}$, $PM_{2.5}$ and $PM_1$ mass concentration measured at the three field sites during the whole campaign period. Top panels show the measured concentrations. Bottom panels shows the mass concentrations that have been normalized to the average mass concentration

Figure 4 provides the mean average diurnal hourly profiles of the $PM_1$, $PM_{2.5}$ and $PM_{10}$ mass concentrations for the three measurement sites during the whole campaign period. There is clear diurnal variation observed at all the sites, two distinct peaks are observed in the two urban locations during the morning (ca. 05:00 – 10:00) and the evening (ca. 18:00 – 24:00) which correspond to the Nairobi peak traffic periods. The normalized data shows that the traffic related structure is very similar in both the urban background and urban roadside sites indicating that the traffic related PM pollution is the dominant source at both sites. The rural background site also shows diurnal variation with some indication of a traffic related signal at similar times to the urban sites, especially in the $PM_{2.5}$ size fraction. However, overall the rural diurnal cycle appears to largely correspond to solar insolation suggesting the dominant factor affecting the rural mass concentrations is the height of the local boundary layer which decreases in the night time and increases with greater solar insolation.

Through comparison of the urban roadside, urban background and rural background hourly averaged data, it is possible to generate estimates of urban increments and roadside increments





relevant for Nairobi using a 'Lenschow' type approach (Lenschow et al., 2001). For the intensive
period the urban and rural increments are calculated for both the $PM_1$, $PM_{2.5}$ and $PM_{10}$ mass
concentrations, see Figure 5.  The urban increment is calculated by subtracting the hourly average
values of the rural background site from the urban background site.  During the intensive period,
analysis of the air mass back trajectories indicates that the regional wind direction was almost
exclusively from the northeast.  Hence the Nanyuki rural background site is a good representative of
the rural background that impacts upon Nairobi.
The roadside increment was calculated by subtracting the hourly average values of the urban
background site from the urban roadside site.  It is noted that the chosen roadside measurement
site is particularly busy with vehicles, compared to many other non-highway streets in Nairobi. In
particular, the site is a popular Matatu (14 seat passenger vans) terminal with multiple vehicles idling
at any point during the day.  Therefore, the roadside increment obtained using this location likely
represents a value close to the upper boundary for Nairobi roads.

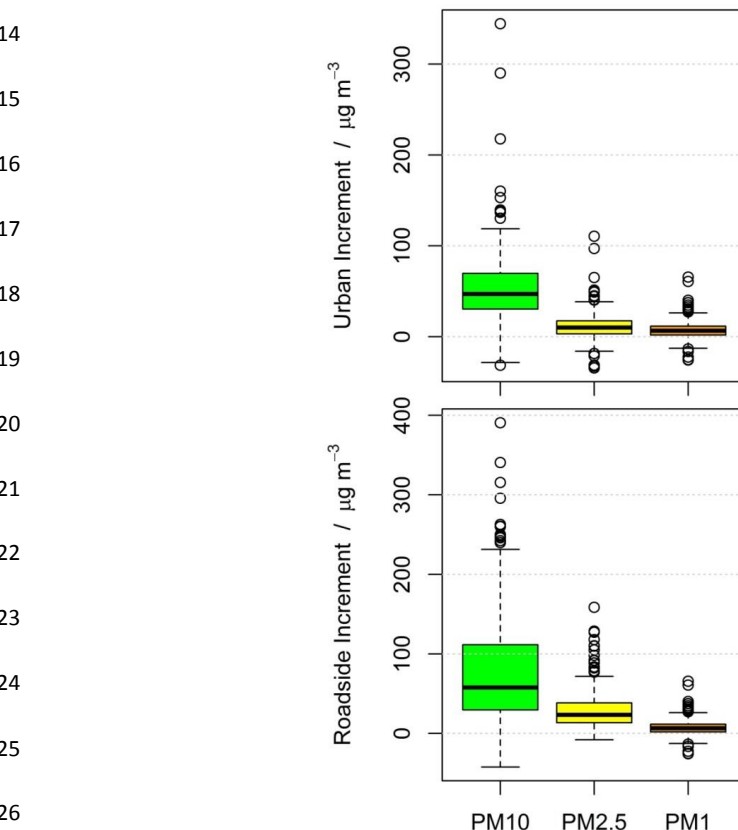





**Figure 5** Box and whisker plots of urban and roadside increment of $PM_{10}$, $PM_{2.5}$ and $PM_1$ calculated
for Nairobi. Data was taken from the intensive campaign period when the urban background, urban
roadside and rural background sites were all measuring simultaneously. Hourly averaged mass
concentration data is used
The urban and roadside increments are significant for all the investigated PM size fractions. A
statistical summary of the roadside and urban increments for the $PM_1$, $PM_{2.5}$ and $PM_{10}$ size fractions
are given in Table 3.
**Table 3** Summary of roadside and urban increments for the $PM_1$, $PM_{2.5}$ and $PM_{10}$ size fractions
measured during the intensive period.

|  | Roadside Increment ($\mu g/m^3$) | | | Urban Increment ($\mu g/m^3$) | | |
|---|---|---|---|---|---|---|
|  | $PM_1$ | $PM_{2.5}$ | $PM_{10}$ | $PM_1$ | $PM_{2.5}$ | $PM_{10}$ |
| Minimum | -4.3 | -6.1 | -31.7 | -20.1 | -26.9 | -23.6 |
| 1st Quartile | 7.3 | 10.5 | 22.2 | 1.0 | 2.2 | 19.5 |
| Median | 12.6 | 18.3 | 43.3 | 4.7 | 7.1 | 33.1 |
| Mean | 18.9 | 22.9 | 58.1 | 5.2 | 8.2 | 36.6 |
| 3rd Quartile | 20.7 | 30.0 | 83.4 | 8.7 | 13.2 | 48.2 |
| Maximum | 95.5 | 123.9 | 292.6 | 51.2 | 86.3 | 258.0 |

During the intensive period, the mean average roadside increment is 57.2, 47.5 and 48.1 % of the
mean roadside mass concentration, in the $PM_1$, $PM_{2.5}$ and $PM_{10}$ size fractions, respectively.
The spatial variation in PM emissions, in the different size fractions, can be assessed at the urban
background and urban roadside sites using bivariate polar plots, which provide information on the
variation of PM mass concentration with wind direction and speed, see Figure 6 (Carslaw and
Beevers, 2013). The urban background and urban roadside sites are sufficiently closely collocated (<
0.5 km apart) that the wind data acquired at the urban background site is applicable to the urban
roadside site. Wind direction data was not available for the rural background site, so analysis of the
spatial variation was impossible at this site.
Figure 6 clearly shows significant variation of PM mass concentration at both urban sites, which are
dependent upon the wind conditions. The urban background site shows broadly similar behaviour in
the spatial variation of the $PM_1$, $PM_{2.5}$ and $PM_{10}$ size fractions. The peak in concentrations are
observed at low wind speeds and when the wind comes from the west and south. This wind
direction dependence is consistent with the close proximity of the major highway A104 'Nairobi-



Malaba Road', which passes close to the site in the direction of high PM concentrations. The diurnal
profiles and roadside increments discussed earlier combined with the wind dependence highlights
the role of roads in Nairobi as the major source of PM in all size fractions studied. Since the site is
within Nairobi's Central Business District (CBD), there are other significant roads nearby as well, but
the A104 has the greatest fleet density.
The urban roadside site also shows distinct variation in pollutant concentrations with wind speed
and direction. In the $PM_{10}$ size fraction the greatest concentrations are seen to the northwest and
smallest to the southwest with a steady reduction between these two extremes. The $PM_{2.5}$ and $PM_1$
size fractions show a more complex behaviour with highest concentrations at low wind speeds and
the north and west directions. The urban roadside location is surrounded by small roads and lower
traffic speeds compared to the highways, for example the A104. The lower traffic speeds likely lead
to less non-tail pipe emissions from dust resuspension and hence there are less local $PM_{10}$ particles
when compared to the urban background site. Whereas the localized $PM_{2.5}$ and $PM_1$ concentrations
are likely due to the heavily congested local roads on which Matatus and other vehicles are often left
idling leading to high tail pipe emissions, which are typically in the smaller PM size fractions (Pant
and Harrison, 2013).

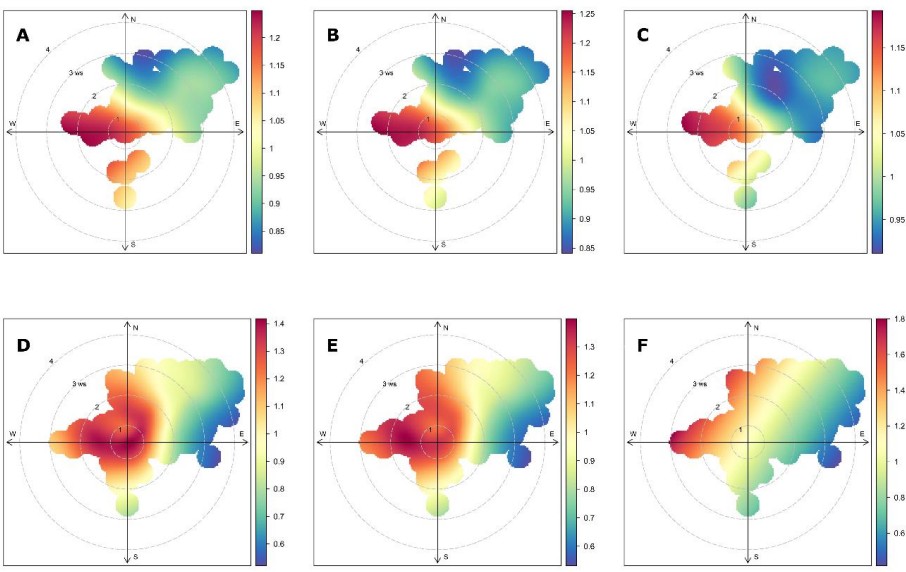



**Figure 6** Bivariate (polar) plots of PM in different size fractions at urban background (UB) and urban
roadside (UR) sites. Panel descriptions A) $PM_1$ UB, B) $PM_{2.5}$ UB, C) $PM_{10}$ UB, D) $PM_1$ UR, E) $PM_{2.5}$ UB
and F) $PM_{10}$ UR. The PM mass concentration data in each plot are normalized to allow for easy
comparison between the different sites and PM size fractions investigated. However, note the scale
bars are different for each panel to allow for easier interpretation.



Figure 7 provides the distributions of the ratio between the coarse and the fine PM mass fractions at
the three field sites during the campaign period. Whilst each site shows distinct variation, with large
interquartile ranges, in the reported coarse:fine ratio, which is dependent on the time of year and
time of day, the median ratios at each site are distinct, with the ratio at the urban roadside, urban
background and rural background sites being 1.6, 1.3, and 0.5, respectively. At the roadside site, the
median coarse:fine mass ratio is almost triple that observed at the rural background; this is
consistent with the dominant source of PM at the roadside site being the resuspension of large dust
particles by non-exhaust emissions from vehicles. At the rural site, the PM size distribution has a
greater ratio of fine material consistent with the rural site having a signature of regional background
PM. The ratio of coarse:fine PM at the urban background site is intermediate between the roadside
and rural background sites which suggests that this site is effected significantly by both the regional
background and the urban road PM sources. These insights into the coarse:fine PM ratio is
consistent with the roadside and urban increments, shown in Figure 5 and discussed previously.

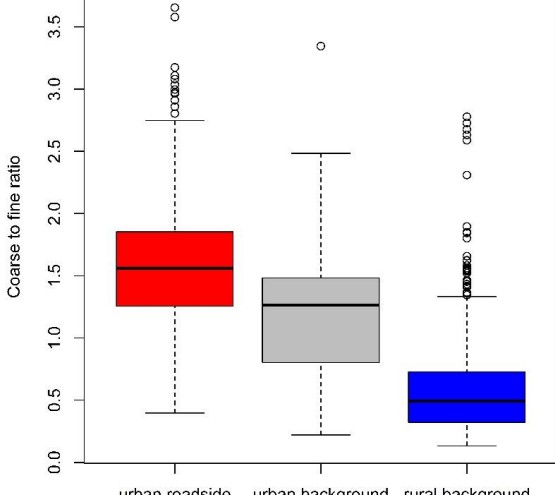

**Figure 7** Ratio of the coarse ($PM_{10} - PM_{2.5}$) to fine ($PM_{2.5}$) fraction of PM
**4.3. Comparison with previous measurements**
To the best of our knowledge, there has been no previous literature study to date utilising calibrated
low cost sensors to measure PM in Nairobi. Furthermore, it is difficult to make comparison with





previous Nairobi based PM studies because of the differences in the temporal resolution of the data
and campaign durations used in this study compared to past measurements.
The most comparable study of $PM_{2.5}$ would be the work of Gaita et al., (2016) which also recorded
the levels of PM $_{2.5}$ at the University of Nairobi (urban background site). The urban background
average of $PM_{2.5}$ during this study's campaign period was 24.8 μg/m$^3$ compared to Gaita et al.,
(2016) mean average of 9.8μg/m$^3$, showing a significant increase of 253%. The sampling time
window used in Gaita et al., (2016) study was between August to September 2007 which is distinct
from the February to March 2017 period of this study. Both of these study periods were largely dry,
with low precipitation levels, thereby suggesting PM deposition would have been similar between
the two studies. The significant increase in measured $PM_{2.5}$ could be due to several reasons.  Firstly,
there could be seasonal differences between August/September and the February/March sampling
periods of the two studies; however, the study of Gaita et al. (2014) suggests the urban background
concentrations of $PM_{2.5}$ mass concentration is similar between these two time periods. The regional
background PM loading may have increased during this time period, potentially due to increasing
regional aridity caused by climate change leading to more dust generation (Greve et al., 2017). There
is almost ten years difference in the times of this study compared to Gaita et al. 2016, in this time
Nairobi has undergone significant increases in population and urbanization with correspondingly
higher use of motorization and fuel.  Using UN population data (UN, 2014), the population of Nairobi
is well modelled by equation E1, in which *Y* is year date, and *p* is the population in thousands, which
suggests that the population of Nairobi has increased by 148% from 2007 to 2017.
$p = 2.33 \times 10^{31} \exp(3.91 \times 10^{-2} \times Y)$          (E1)
Hence, the population increase alone cannot account for the increase in PM concentration.  The
pollution production capability per capita could have increased, which is very likely because of the
increased rates of motorization and fuel use.  If we assume that the increase in PM is solely due to
population increase and per capita pollution, it suggests that in 2017 the average citizen is 70% more
polluting than the average citizen in 2007.
The Egondi et al. (2016) study of $PM_{2.5}$ in two slums in Nairobi reported much higher values of 166
μg/m$^3$ and 67 μg/m$^3$ for two different slum areas within Nairobi. These values are much higher than
the average $PM_{2.5}$ values from this study; Egondi et al. stated that the reason for such high levels of
$PM_{2.5}$ stemmed from the local situation and distinct sources of PM within the two slums. This study
used a TSI optical particle counter, which was placed 1.5 m above ground level. Therefore, these
measurements were likely highly influenced by re-suspended dust.



Although Kinney et al., (2012) measured PM$_{2.5}$ levels at four roadside locations, the sampling
window was only 11 hours and therefore it is not possible to directly compare this study to it.
However, considering the diurnal variation in PM found in this study, both investigations measured
similar PM$_{2.5}$ levels. Kinney et al., (2012) recorded daytime concentration ranges of 10.7 µg/m$^3$ and
98.1 µg/m$^3$ for a rural and urban roadside site, respectively, compared to ca. 25 µg/m$^3$ and ca. 150
µg/m$^3$ for this study. Again, the increase between sampling years may be a reflection of the
increased population, vehicular traffic and rapid urbanisation.
**5. Discussion**
In this study, we have shown that Nairobi currently has very high levels of PM mass concentration in
the PM$_1$, PM$_{2.5}$ and PM$_{10}$ mass fractions. These measurements were conducted using low cost
calibrated OPC-N2 sensors. The measured PM$_{2.5}$ and PM$_{10}$ concentrations at the urban roadside and
urban background sites both regularly exceeded the WHO daily limits and very likely exceed the
annual limits. In particular, the roadside site often showed concentrations of double the WHO
guidelines. These concentrations will very likely be causing significant harm to the population of the
Nairobi.
The negative health effect of PM is linked to the level of exposure experienced by the patient. This
paper and others (e.g. Gaita et al. 2014) have shown that in Nairobi, vehicle emissions are the most
significant source of PM. Hence, in Nairobi and other similar cities, the exposure to outdoor PM is to
a large extent a function of ones proximity to roads. Furthermore, since traffic varies diurnally,
seasonally and by day of the week, personal exposure is both spatially and temporally dependent.
This spatial and temporal heterogeneity leads to health inequalities in cities.  The urban poor who
are often most vulnerable to environmental risks due to lack of adequate health provision, typically
live in close proximity to roadways, heightening their exposure to vehicular emissions. Stemming
from poorly planned rapid urbanisation and inadequate service provision within these cities, those
that are unable to afford public transport or personal vehicles frequently walk along these pollution
heavy roads, only increasing their exposure periods.
This study only looked at outdoor air quality, it is important to stress that most air pollution deaths
in Kenya and SSA in general are due to poor indoor air quality.  As a total number of deaths, deaths
related to indoor air quality in Kenya rose to 18% from 1990-2013 (Roy, 2016).  In LMIC countries,
indoor exposure to pollutants is typically from the household combustion of solid fuels on open fires
or traditional stoves. These exposures increase the risk of acute lower respiratory infections and
associated mortality among young children; indoor air pollution from solid fuel use is also a major





risk factor for cardiovascular disease, chronic obstructive pulmonary disease and lung cancer among
adults (Muindi et al., 2016).
This study has shown that the low cost OPC-N2 sensors can be used to generate diurnal PM datasets
with good precision and repeatability. As noted in the methodology, it would have been preferable
for more cross calibration periods with the tried and tested gravimetric PM measurement but
resources did not allow this.  In addition to more calibration points, the study could have been
enhanced by the inclusion of collocated calibration points for the roadside and rural background
sites in addition to the urban background site, since the average particle shape, size and density will
likely be different between the three sites because of differing PM sources and emission factors.
However, it is noted, that whilst it is desirable from a purely scientific point of view to have more
inter-comparison with reference grade equipment; every inter-comparison adds significant
additional cost to the project both in terms of consumables for the gravimetric analysis (including
the cost of analytical grade filters and accompanying laboratory supplies), and the cost in
manpower. Many other cities in SSA and other LMIC countries do not have the resource that Nairobi
does in having a gravimetric sampler. These additional costs required for highly accurate scientific
results would likely make the low cost sensors not so very low cost after all, and hence bring into
question their unique selling point (USP).
Whilst this paper focused on PM pollution, it is noted, that there are serious risks to health not only
from exposure to PM, but also from exposure to ozone ($O_3$), nitrogen dioxide ($NO_2$) and sulfur
dioxide ($SO_2$). As with PM, concentrations are often highest largely in the urban areas of low- and
middle-income countries. Ozone is a major factor in asthma morbidity and mortality, while nitrogen
dioxide and sulfur dioxide also can play a role in asthma, bronchial symptoms, lung inflammation
and reduced lung function.  Good quality measurements of these gas phase pollutants lag behind
measurements of PM in Nairobi, other SSA cities and LMIC cities in general. This is due to the high
importance of PM as an environmental risk factor but also because of the lack of good quality gas
analysers which are affordable and transportable.
**6. Conclusions**
Air quality in many LMIC urban centres is often poor and in many cities is getting worse due to the
combined pressures of increasing population, increasing urbanization, increasing vehicular traffic
and poor vehicle regulation. To be able to manage air pollution, good quality and long term data sets





are required. Unfortunately, in many LMICs the cost of certified high quality air quality
measurements is beyond the financial means of environmental authorities. Low cost sensors offer
the possibility of air quality products at significantly lower cost compared to traditional methods.
This paper used calibrated OPC-N2 devices to measure PM concentrations in Nairobi, Kenya in the
size fractions $PM_1$, $PM_{2.5}$ and $PM_{10}$. The data required calibration using an established gravimetric
approach to PM measurement. The need for calibration by trained scientists significantly increases
the costs associated with low cost monitoring and this cost needs to be factored in when assessing
options for air quality monitoring.
PM was measured in three locations: an urban roadside, urban background and rural background
site for a period of approximately two months. The data reveals that roadside and urban
background locations in Nairobi often exceed the WHO guidelines for daily averaged PM mass
concentration in both the $PM_{2.5}$ and $PM_{10}$ size fractions. Comparison of the data with previous
measurements conducted in Nairobi is difficult but where comparison is possible, it appears that air
quality has become worse in the last ten years which is likely due to increases in population,
urbanisation and motorization. Comparison of the data from the three sites, following a 'Lenschow'
type approach, allowed for the calculation of representative roadside and urban increments for
Nairobi (Lenschow et al., 2001). This increment data can be used in future air quality modelling to
assess the likely health impact of PM pollution on Nairobi's population. The combination of the
diurnal PM data with local meteorology allows for simple source apportionment of the PM. The
diurnal PM concentrations tracks the Nairobi rush hours, furthermore, PM peaks when the wind
comes from the direction of significant numbers of vehicles such as major roads and a Matatu stop.
These facts taken together, point towards vehicle emissions being the major sources of air pollution
in Nairobi, as has been previously observed in studies such as Gaita et al. 2014. The coarse PM
fraction increases at roadside compared to urban background site suggesting that non-exhaust
vehicle emissions make up a significant amount of the vehicle emissions.
In summary, the low cost sensors used in this study provided much useful data for assessing air
quality in Nairobi at an equipment cost significantly lower than that of traditional instruments. Low
cost sensors have great potential in other country settings and could be used for long term sampling
if the appropriate calibrations are performed.
**Acknowledgements**
Leigh Crilley and Robin Price are thanked for their help in optimizing the OPC-N2 monitoring
systems. This work was funded via an EPSRC grant (Global Challenges Research Fund IS2016), the
Royal Society and Royal Society of Chemistry International Exchanges Award (IE170267), and DFID





via the East African Research Fund (EARF) grant 'A Systems Approach to Air Pollution (ASAP) East
Africa'. Scientific research support by International Science Programme in Sweden to Institute of
Nuclear Science & Technology in University of Nairobi was appreciated.

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
