# Peer review of "Manuscript under review for journal Atmos. Chem. Phys."

_Atmospheric Chemistry and Physics, 2018_

## Referee Comment (RC1) · Anonymous Referee #1 · 28 May 2018

I think the data presented in this manuscript is important. I also think the author's efforts to get as many insights from the data is very good. However, I have concerns about the calibration methodology. Although the authors themselves point out the concerns, I think the manuscript needs to specify how the OPC works in more detail and speak more about the validity of this calibration. Is a simple linear fit okay? How does one take into consideration the different aerosol size distributions and types at the other locations and wouldn't that influence the calibration dramatically? Is it worthwhile calibrating the OPCs in the urban background site in the first place if you're going to use the OPCs at other sites? What does the literature say about this? I appreciated the discussion on the RH and its impacts on measured PM, but I wonder about other aerosol properties: shape composition that must be mentioned here

[Figure]

I'd also like to see an image of the OPC if possible. Is it pole mounted etc?

---

## Referee Comment (RC2) · Anonymous Referee #2 · 6 Jun 2018

Major Comments: As the authors state, there are few PM measurements in Africa, thus the data presented here are important. Additionally, the use of low-cost monitors is of growing interest and information on the use of these instruments is beneficial to the field. However, the organization and analysis in the paper could be improved.

One of the main issues with the paper is the use of "calibrated" and the authors purporting that it is a major strength of the study. I am not entirely convinced of the authors' calibration methods. While the authors acknowledge the limitations in having only one day at one location; I'm not totally convinced that the calibration even improves the results. A scaling factor determined from one day (with results that have a pretty large uncertainty range) cannot represent the variability in aerosol size distributions, composition, or relative humidity that might impact the results. They mention these differences

when comparing their results to a previous study in the UK, but then assume it does not make a difference between their sites. The authors should just be more cautious in stating that they calibrated the results and not overstate the significance of the calibration (since they did not actually test that the calibration improves their results).

I am also a little confused by the "Lenschow" increment section. The authors separate out an urban background from an urban roadside increment. What do these increments actually represent and what is the bigger implication? Through most of the paper, they discuss urban emissions as primarily vehicle emissions and the major source for the urban background site seems to be the highway. In the Conclusion section, they say these could be useful for modeling studies, but I am unsure of how since it is not clear what they represent.

Additionally, the introduction is much too long but could benefit from being trimmed down. The extensive literature review on all previous measurements does not seem necessary, and the information is repeated again in sections 4.3 and 5.

Please increase the font size on all the figures.

Finally, there is a lack of citations in some parts of the paper or strange choices in citations (noted below), along with some odd word choices throughout that I think are more literary in style than necessary (examples: whilst, henceforth, fortnight, vanguard, bespoke).

Minor comments: Page 1, Line 11: change to "study provides much needed"

Page 1, Line 20: what is "fraction"? is this an actual fraction or the PM2.5 mass concentration?

Page 1, Line 29: "Lenschow type approach" needs a citation.

Page 1, Line 31-Page 2, Line 2: "Respectively" is used three times in this sentence alone. In general, "respectively" is overused in this paper.

Page 2, Lines 10-11: the sentence "The potential problems…" seems out of place. I would remove it.

Page 2, Line 17: "attributed" should be "contributed" or "1 in 4 deaths is attributable to …"

Page 2, line 28: remove "air pollution"

Page 2, Line 29: citation should be e.g. and this study only looked at long-term exposure and mortality so it does not apply to the whole statement. Also, what are "short term effects on human mortality"?

Page 2, Line 31: I do not think this is the best citation. I think there are a lot of journal articles that would be better references.

Page 3, Line 1: This does not need a citation.

Page 3, Lines 11-13: need a citation

Page 3, Line 28: Nairobi is in Africa, so just put "in Africa"

Page 4, Line 11: Please remove this sentence or rewrite it, as is it is not true.

Page 4, Line 32 and Page 13, Line 3: circa is generally used for dates, not measurements.

Page 5, Lines 1-2: change to "could be a significant health concern"

Page 7, Lines 19-23: This is not really methodology and should be left to the introduction or put in the discussion section.

Page 8, Line 20: change to "was mounted about 4 m"

Page 10, Line 3: Remove "The AlphaSense…OPC-N2" as it is already referred to in the parentheses of the previous sentence.

Page 10, Lines 14-16: The authors are using firmware version 18, so what is the

additional weighting?

Page 11, Lines 31-32: The OPC measurement does not have an uncertainty range.

Page 11, Lines 32-33: Did the authors determine these uncertainties for the gravimetric concentration or are these from the literature?

Page 13, section 3.5 This seems out of place in the methodology section. I would perhaps shorten this section and put it in with the discussion section.

Page 13, Line 7: remove "of the Earth". I would also suggest pointing out that this is from a model.

Page 13, Line 10: remove "derived"

Page 13, Lines 10-12: Is there a citation for this? I think of this as true for many regions because of aging downwind of urban area making aerosols more hygroscopic, but I am not sure about this for Africa. What do the authors think is the composition of the rural/regional background vs. the urban?

Page 13, Lines 18-20: There may not appear to be a dependence from the plot because there is so much scatter. However, their assertion depends on the assumption that all these aerosols are the same and experiencing different RH levels. Potentially subsetting the data for like aerosols would show a dependence. The authors should just be less emphatic that there is no dependence. Also, aerosols take up water at relative humidity values less than 85%. The uptake will depend on the composition as the authors mention, so I am not entirely sure that a study completed with a completely different aerosol type should negate the potential effect for this study and would therefore suggest the authors not rely so much on the "85% threshold" for their comparisons.

Section 4.1 This can all go in the supplement.

Figure 2: Use a legend rather than the caption to explain the figure lines

Page 15, Lines 7-11: This seems more like methodology as compared to results.

Page 16, Lines 8-12: This seems like a discussion point and could use more proof that it is long range pollution (could be a regional event?).

Page 16, Line 17-Page 17, Line 10: I do not think calculating an annual average from 25-40 days of measurements in one season is useful. This section should be removed.

Figure 3: These are hourly concentrations. It does not make sense to add on the annual and daily WHO guidelines. Should make a separate plot with the daily averages.

Page 19, Lines 17-20: There is no plot of solar insolation, so just say that it is likely affected by the boundary layer height.

Figure 6: Can the labels be put on the actual plot rather than just in the caption?

Page 24, Line 9: Remove "non-exhaust emissions from vehicles"

Page 25, Lines 25-26: I am not sure that this is a good calculation to even suggest. The authors suggested that the highway was a major source for the urban background. The highway runs through the city, suggesting that traffic through the city, not changes in the urban population would be a major driver of the increasing pollution.

Section 5. I don't know if this needs to be its own section. It should either be put in the Results or in the Conclusion as quite a bit of it is simply a repeat.

Page 26, Lines 6-7: Any changes in industry?

Page 26, Lines 21-26: Need citations.

Page 26, Lines 27-28. Needs a citation.

Page 26, Lines 29-31. Needs a citation.

Page 27, Lines 19-27: Need citations.

---

## Author Comment (AC1) · 12 Aug 2018

**Airborne particulate matter monitoring in Kenya using calibrated low cost sensors**

Response to all reviewer's comments

We thank the reviewers for taking their time to comment on this paper. Their suggestions have improved the paper. We now go through each comment one by one, responses in red.

Anonymous Referee #2

**Major Comments:**

1) As the authors state, there are few PM measurements in Africa, thus the data presented here are important. Additionally, the use of low-cost monitors is of growing interest and information on the use of these instruments is beneficial to the field.

Response: We are happy that the reviewer sees the value in the work.

2) The organization and analysis in the paper could be improved.

Response: in replying to the various comments we have improved the organization and the analysis, see specific responses below.

3) One of the main issues with the paper is the use of "calibrated" and the authors purporting that it is a major strength of the study. I am not entirely convinced of the authors' calibration methods. While the authors acknowledge the limitations in having only one day at one location; I'm not totally convinced that the calibration even improves the results. A scaling factor determined from one day (with results that have a pretty large uncertainty range) cannot represent the variability in aerosol size distributions, composition, or relative humidity that might impact the results. They mention these differences when comparing their results to a previous study in the UK, but then assume it does not make a difference between their sites. The authors should just be more cautious in stating that they calibrated the results and not overstate the significance of the calibration (since they did not actually test that the calibration improves their results).

Response: in writing the paper we were very careful not to oversell and make clear in several places where the study could have been improved if time and finances had allowed. In the abstract, we state that calibration only occurred at one site. In section 3.4, we clearly state the date and location of the calibration. The following statement makes clear that the calibration at the non-urban background sites should be treated more carefully: "The gravimetric calibration was carried out at the urban background field location, for the three OPC-N2s which were subsequently used in the measurement campaign at the three field sites. Hence, the calibration was most appropriate for the urban background site. Whilst the urban roadside site is in close proximity to the urban background site, the roadside site is more influenced by traffic related PM, hence, the average particle density at the roadside site is likely different to the urban background site. Likewise, the rural background site is

likely to be far more influenced by mineral dust than the two urban sites. Hence the gravimetric calibration at the urban background sites only provides an estimate calibration for the urban roadside and rural background sites. Only one gravimetric calibration was carried out during the study period due to the lack of resource for further calibrations. If the PM composition varied significantly over the study period, then the true calibration factor will also change. Hence, the calibration factor used should be treated as an estimate for the whole study period because changes in PM composition lead to changes in particle refractive index, and therefore, the scattering pattern which is measured by the OPC to estimate particle size. Changes in particle density, due to compositional changes, also affects the particle mass calculated from the particle size. It is noted, for future studies it would be beneficial to have multiple gravimetric calibration points to check for continuing accuracy of the OPC-N2 sensors throughout the campaign."

4) I am also a little confused by the "Lenschow" increment section. The authors separate out an urban background from an urban roadside increment. What do these increments actually represent and what is the bigger implication? Through most of the paper, they discuss urban emissions as primarily vehicle emissions and the major source for the urban background site seems to be the highway. In the Conclusion section, they say these could be useful for modeling studies, but I am unsure of how since it is not clear what they represent.

Response: the "Lenschow" approach is a widely used approach to generate a first order estimate of air pollution within a city. The difference between the urban background site and the rural background site represents an estimate of the minimum exposure to air pollution anywhere in the city. The urban background site is chosen so it is removed from any localised source of pollution and is hence representative of city wide pollution. Since air pollution has a significant vehicular component in Nairobi, it is expected that the urban increment will show significant vehicular component. The following text has now been placed in the introduction "The Lenschow approach allows for simple modelling of urban air pollution based on the urban and roadside increments in air pollution".

5) Additionally, the introduction is much too long but could benefit from being trimmed down. The extensive literature review on all previous measurements does not seem necessary, and the information is repeated again in sections 4.3 and 5.

Response: we believe the long format of ACP allows this level of detail and it is useful for the reader.

6) Please increase the font size on all the figures.

Response: done.

7) Finally, there is a lack of citations in some parts of the paper or strange choices in citations (noted below), along with some odd word choices throughout that I think are more literary in style than necessary (examples: whilst, henceforth, fortnight, vanguard, bespoke).

Response: responses given on a case by case basis below.

**Minor comments:**

1. Page 1, Line 11: change to "study provides much needed" Changed
2. Page 1, Line 20: what is "fraction"? Is this an actual fraction or the PM2.5 mass concentration? Changed to "fine particle fraction ($PM_{2.5}$)".
3. Page 1, Line 29: "Lenschow type approach" needs a citation. Changed
4. Page 1, Line 31-Page 2, Line 2: "Respectively" is used three times in this sentence alone. In general, "respectively" is overused in this paper. Change to "The median urban increment is 33.1 µg m$^{-3}$ and the median roadside increment is 43.3 µg m$^{-3}$ for $PM_{2.5}$. For $PM_1$, the median urban increment is 4.7 µg m$^{-3}$ and the median roadside increment is 12.6 µg m$^{-3}$.
5. Page 2, Lines 10-11: the sentence "The potential problems. . ." seems out of place. I would remove it. Calibration of low cost sensors is required, we believe this line should be kept.
6. Page 2, Line 17: "attributed" should be "contributed" or "1 in 4 deaths is attributable to . . ." changed
7. Page 2, line 28: remove "air pollution" changed
8. Page 2, Line 29: citation should be e.g. and this study only looked at long-term exposure and mortality so it does not apply to the whole statement. Also, what are "short term effects on human mortality"? People who are already susceptible to underlying respiratory disease, pneumonia, influenza or asthma can see a worsening of their symptoms and illnesses through short term exposure to air pollution (https://doi.org/10.1016/j.atmosenv.2012.10.019)
9. Page 2, Line 31: I do not think this is the best citation. I think there are a lot of journal articles that would be better references. Reference changed to recent Lancet report
10. Page 3, Line 1: This does not need a citation. Removed
11. Page 3, Lines 11-13: need a citation Citation provided
12. Page 3, Line 28: Nairobi is in Africa, so just put "in Africa" Changed
13. Page 4, Line 11: Please remove this sentence or rewrite it, as is it is not true. Changed
14. Page 4, Line 32 and Page 13, Line 3: circa is generally used for dates, not measurements. Changed to approximately
15. Page 5, Lines 1-2: change to "could be a significant health concern" Changed
16. Page 7, Lines 19-23: This is not really methodology and should be left to the introduction or put in the discussion section. We believe this information is relevant to methodology and have left unchanged.
17. Page 8, Line 20: change to "was mounted about 4 m" Changed
18. Page 10, Line 3: Remove "The AlphaSense. . .OPC-N2" as it is already referred to in the parentheses of the previous sentence. Changed
19. Page 10, Lines 14-16: The authors are using firmware version 18, so what is the additional weighting? The Alphasense manual does not provide any further information other than what is stated already in the manuscript.
20. Page 11, Lines 31-32: The OPC measurement does not have an uncertainty range. The value is as provided by the OPC, it is not possible to provide an error from the 1 day calibration. From the Crilley et al. (2018), the coefficient of variance is estimated as 0.32 ± 0.16, 0.25 ± 0.14 and 0.22 ± 0.13 for PM1, PM2.5 and PM10 mass concentrations, respectively. This is now stated in the updated manuscript.
21. Page 11, Lines 32-33: Did the authors determine these uncertainties for the gravimetric concentration or are these from the literature? The uncertainty in gravimetric concentrations was estimated from the instrument (10%), sampling (7%) and weighing (25%) errors. This information is now included in the manuscript.

22. Page 13, section 3.5 This seems out of place in the methodology section. I would perhaps shorten this section and put it in with the discussion section. This section answers major comment 2.

23. Page 13, Line 7: remove "of the Earth". I would also suggest pointing out that this is from a model. Sentence changed to "The average κ parameter values for Africa (κ = 0.15±0.12) are lower than for Europe (κ = 0.36±0.16), as based on the Pringle et al. 2010 model, which is in good agreement with observational data"

24. Page 13, Line 10: remove "derived" Changed to "sourced from"

25. Page 13, Lines 10-12: Is there a citation for this? I think of this as true for many regions because of aging downwind of urban area making aerosols more hygroscopic, but I am not sure about this for Africa. What do the authors think is the composition of the rural/regional background vs. the urban? We found no good citations for African particle hygroscopicity and its link to chemical aging. However, on reappraisal the sentence was too strong without supporting evidence. Correspondingly, the sentence is toned down by changing it to "However, PM derived from urban emissions are often less hygroscopic than rural PM; therefore, the rural estimates might provide a useful upper estimate of particle hygroscopicity in urban centres."

26. Page 13, Lines 18-20: There may not appear to be a dependence from the plot because there is so much scatter. However, their assertion depends on the assumption that all these aerosols are the same and experiencing different RH levels. Potentially subsetting the data for like aerosols would show a dependence. The authors should just be less emphatic that there is no dependence. Also, aerosols take up water at relative humidity values less than 85%. The uptake will depend on the composition as the authors mention, so I am not entirely sure that a study completed with a completely different aerosol type should negate the potential effect for this study and would therefore suggest the authors not rely so much on the "85% threshold" for their comparisons. We see no dependence of RH on aerosol mass in the data presented in this paper. Even taking into account the scatter in the plot, there is no suggestion of an RH effect over the RH range measured. We agree that different aerosol compositions might show different relationships but since we have no methodology for subsetting the data into individual aerosol compositions, we cannot do this analysis.

27. Section 4.1 This can all go in the supplement. Changed and changed table numbers/contents numbers

28. Figure 2: Use a legend rather than the caption to explain the figure lines We have added a caption as requested.

29. Page 15, Lines 7-11: This seems more like methodology as compared to results. We believe this section makes sense where placed.

30. Page 16, Lines 8-12: This seems like a discussion point and could use more proof that it is long range pollution (could be a regional event?). This is beyond the scope of the paper, we mention the possibility of a long range event as a possible explanation. We lessen the strength of the statement by stating "this might represent" rather than "it likely represents".

31. Page 16, Line 17-Page 17, Line 10: I do not think calculating an annual average from

32. 25-40 days of measurements in one season is useful. This section should be removed. Useful as there is no dataset like it and has been clearly stated that it is estimated

33. Figure 3: These are hourly concentrations. It does not make sense to add on the annual and daily WHO guidelines. Should make a separate plot with the daily averages.

34. Page 19, Lines 17-20: There is no plot of solar insolation, so just say that it is likely affected by the boundary layer height. We believe statement already uses caution by using "suggesting" rather than anything more definite.

35. Figure 6: Can the labels be put on the actual plot rather than just in the caption? Changed.

36. Page 24, Line 9: Remove "non-exhaust emissions from vehicles" Changed

37. Page 25, Lines 25-26: I am not sure that this is a good calculation to even suggest. The authors suggested that the highway was a major source for the urban background. The highway runs through the city, suggesting that traffic through the city, not changes in the urban population would be a major driver of the increasing pollution. This is an interesting question. Whilst traffic flows will unlikely be exactly linear with population, they are clearly related. We think this analysis is interesting to the field and is described in an honest way that points out where flaws may exist in its logic, "If we assume that the increase in PM is solely due to 25 population increase and per capita pollution…"

38. Section 5. I don't know if this needs to be its own section. It should either be put in the Results or in the Conclusion as quite a bit of it is simply a repeat. we believe the long format of ACP allows this level of detail and it is useful for the reader.

39. Page 26, Lines 6-7: Any changes in industry? This is beyond the scope of the paper, but the possibility is now included by adding the following sentence "It is noted, changes in industry may also influence the air quality".

40. Page 26, Lines 21-26: Need citations. Done

41. Page 26, Lines 27-28. Needs a citation. Done

42. Page 26, Lines 29-31. Needs a citation. Done

43. Page 27, Lines 19-27: Need citations. Done

Anonymous Referee #1

1) I think the data presented in this manuscript is important. I also think the author's efforts to get as many insights from the data is very good. However, I have concerns about the calibration methodology. We are happy that the reviewer sees the value in the work.

2) Although the authors themselves point out the concerns, I think the manuscript needs to specify how the OPC works in more detail and speak more about the validity of this calibration. Please see response to Anonymous Referee #2 question 3.

3) Is a simple linear fit okay? For the calibration approach taken, only a linear fit is possible.

4) How does one take into consideration the different aerosol size distributions and types at the other locations and wouldn't that influence the calibration dramatically?

5) Is it worthwhile calibrating the OPCs in the urban background site in the first place if you're going to use the OPCs at other sites? We believe so. The OPCs are factory calibrated in the UK under UK conditions. Whilst the urban background site in Nairobi is not the same as the rural background site and Nairobi urban roadside site, it is likely to have more similar conditions compared to the UK calibration conditions.

6) What does the literature say about this? The literature on low cost sensors is still nascent an only just finding its feet with respect to calibration. Our Crilley et al. (2018) paper suggests it is best to calibrate wherever the low cost sensor is mounted. However, to calibrate everywhere would take significant resources and would put into question the 'low cost' aspect of the 'low cost sensor'.

7) I appreciated the discussion on the RH and its impacts on measured PM, but I wonder about other aerosol properties: shape composition that must be mentioned here. These are al important parameters, but there was no possibility to measure them.

8) I'd also like to see an image of the OPC if possible. Is it pole mounted etc? A photograph of the sensor package is now provided in the supplementary material. The OPC were mounted to railings allowing for "The sensor boxes were placed in locations free 27 from obstacles, at

the three measurement sites, allowing for 360 degrees of air flow", as stated in the manuscript.